# MatriXSSed: A New Taxonomy for XSS in the Modern Web

## Abstract

Cross-site scripting (XSS) constantly remains one of the most prevalent attacks on the Web. In this work, we question its current taxonomy, i.e., the client- or server-side reflected (non-persistent) or stored (persistent) matrix. The Web has extensively changed. Consequently, considering XSS with the lenses of this famous matrix has become at least imprecise, at most impossible for many code injection scenarios where (i) a service worker or an edge worker generates HTTP responses and can reflect or persist XSS payloads infecting not only JavaScript in web pages but also Web assembly, web workers and affecting one or many users automatically; (ii) an attacker sends a web push message directly to a browser push service to trigger code execution in a dormant service worker; or (iii) a cross-origin adversary tampers with code stored by a vulnerable website on the user's physical/permanent file system, etc. Our proposal –to get out of the matrix and not enter another rigid one– expresses the essence of XSS as code infection and affection attack, and allows for clearly specifying the different actors and components involved, their environments, contexts and storages, as well as their recurrence and persistence seen as a continuum rather than a binary marker. From a defensive perspective, we showcase the challenges and limitations of current mechanisms at mitigating XSS targetting the entire attack surface of modern websites. Finally, we demonstrate an abuse of the *Service-Worker-Allowed* (SWA) header to control entire domains with malicious service workers.

**ACM Reference Format:**
Anonymous Author(s). 2025. MatriXSSed: A New Taxonomy for XSS in the Modern Web. In *Proceedings of The Web Conference 2025* . ACM, New York, NY, USA, 10 pages. https://doi.org/XXXXXXX.XXXXXXX

## 1 Introduction

The Web operates on a foundational client-server architecture and a set of principles that enable seamless access and interaction with resources across the globe while shaping the way the security community understands and approaches it. At its core, HTML (HyperText Markup Language) [13] defines how to structure and render web pages (documents). With their *linking*, *embedding* or *hosting* features, HTML documents can reference other resources, in particular, CSS (Cascading StyleSheets) for the presentation, and JavaScript for interactivity. The HTTP (HyperText Transfer Protocol) defines the rules and formats for data exchange and communications between clients, and servers connected to the Internet.

| Server | Stored | XSS | Client | Stored | XSS |
|--------|--------|-----|--------|--------|-----|
| Server | Reflected | XSS | Client | Reflected | XSS |

**Figure 1: Types of XSS attacks according to the OWASP [21]**

URLs (Uniform Resource Locators) serve as the addresses that identify the location of resources. The derived concept of origin (e.g., *https://www.example.com*) is defined as a combination of the protocol (e.g., *https*), domain (e.g., *www.example.com*), and port (e.g., *443*). The same-origin policy (SOP) principle, built on this concept, prevents unauthorized access to Web data across different (cross-) origins while allowing same-origin interactions in most cases. Web browsers have imposed themselves as the most popular clients or user agents, implementing and offering the necessary technological stack, for users to smoothly interact with the Web. Since its conceptualization in 1999, XSS (cross-site scripting) has remained one of the most prevalent attacks. Figure 1 presents the current XSS taxonomy as a matrix adopted by the OWASP (Open Worldwide Application Security Project) and the research community as a reference to frame all types of XSS, which can be server-side, or client-side stored (persistent) or reflected (non-persistent). From a defense perspective, input sanitization, context-aware output encoding, web application firewalls, and the content security policy (CSP) can be deployed to mitigate XSS [36].

In more than two decades, the Web has extensively changed, and in particular, its attack surface which has largely expanded with the numerous new technologies and features. Notably, Web assembly (Wasm) provides an additional code execution environment to complement the pervasive JavaScript. Dedicated, shared, and service workers, and more recently, worklets (lightweight workers) can be spawned to execute code in dedicated threads, independently of the main UI thread where HTML content is rendered and interacted with by users and manipulated by JavaScript via the DOM (Document Object Model) API. Service workers in particular bring various privileged features like the fetch event which empower them with proxy-like capabilities and the ability to intercept HTTP requests and respond to them, or receive web push messages in the background, independently of whether the user is actively interacting with the website or not. Conceptually at the middle of the network sit content delivery networks (CDNs) and other serverless edge computing services which promise geographical proximity to users, and cache and route content quickly where and when it is needed. Vendors like Cloudflare or Akamai are extensively popular among websites [4]. A notable recent client-side storage mechanism is the file system API which brings file management capabilities to websites [10]. Specifically, its file system access extension gives websites direct read and write accesses into the user's physical file system, without SOP or cross-browser restrictions [1]. Figure 2 shows these components positioned in an example architecture.

*Does the current taxonomy of attacks and defenses properly frame XSS in the progressive web, considering the advent of features like (service) workers, serverless web cache proxies, Wasm, or permanent*

*cross-origin storage mechanisms like the file system?* This is the research question we propose to tackle with this study. We note that even though the Web has drastically changed, its attack surface extensively expanded, and numerous important features introduced, XSS is still largely framed from the lenses of browsing contexts, i.e., web pages. To fill this gap, we focus in this work on the other contexts and environments and discuss their specificities. From an overview of XSS, its threat model, and defenses, we discuss the limitations of the current taxonomy and defenses (Sections 3 and refsec:defenses), in light of the new features and the complex Web architecture (Section 2). Then, we propose a step forward in revising this taxonomy to account for the state of the modern Web going forward (Sections 4 and 5). Specifically, by sticking to the essence of XSS as a code (i.e., special content) injection attack, we propose a descriptive frame for clearly specifying the different components and actors (user or attacker) involved in the malicious payloads delivery process (infection), the code execution phase (affection), as well as the *recurrence* of exploits gained by the attacker with a campaign. Furthermore, considering that code execution environments, contexts, and storage mechanisms can all have different levels of persistence, we propose to frame the latter as a continuum instead of a binary marker (persistent vs. non-persistent). We then demonstrate how to express prior XSS attacks and the ones we discussed in this work (See the Appendix). Finally, we demonstrate how a peculiarity of service workers via the *Service-Worker-Allowed* header can be abused to take control of entire domains in shared hosting setups (Section 6).

| API | Description |
|---|---|
| `WebAssembly.instantiateStreaming` `WebAssembly.instantiate` `WebAssembly.complieStreaming` `WebAssembly.compile` | Web Assembly compilation and execution |
| `navigator.serviceWorker.register` | Service worker registration |
| `PushManager.prototype.subscribe` | Subscribe to web push notifications |
| `new Worker` | Create dedicated workers |
| `new SharedWorker` | Create shared workers |
| `Worklet.prototype.addModule` | Spawn worklets |
| `showDirectoryPicker` `showOpenFilePicker` `showSaveFilePicker` | User's physical file system access API |
| `navigator.storage.getDirectory` | The Origin Private System API |
| `FileSystemHandle` `FileSystemFileHandle` `FileSystemDirectoryHandle` | The File System API interface |

**Table 1: Excerpt of Modern Web Features**

## 2 Background

In the Web, HTML hosts JavaScript, which in turn can host Web Assembly (Wasm), worklets, and workers. Workers can further load Wasm and workers, etc. Table 1 presents the related JS APIs.

### 2.1 Web Assembly (Wasm)

With Wasm, the Web enjoys an additional environment that executes compiled code originally developed in low-level programming languages like C/C++ or Rust. Wasm can be represented in textual forms suitable for human inspection, and displayed in text editors and web browsers to assist in debugging. It can be compiled and

executed in environments as diverse as the Cloud, and serverless edge applications, either as standalone programs or integrated into other environments. Notably, in browsers JS and Wasm are executed in different runtime environments. Specifically, Wasm runs in a standalone contained environment, without access to same-origin features like client-side storage, or communication APIs like fetch. To interact, there is a mechanism of import (from JS to Wasm) and export (from Wasm to JS) the different environments can leverage to expose an interface, e.g., objects and in particular functions that can be called cross-environments. Listing 1 shows a code snippet that imports Wasm and exposes a set of functions for it to call.

```
WebAssembly.instantiateStreaming(fetch("/wasm.wasm"), {
    jsapis: function(func, args) {
        if (self[func]) self[func](...args)
    },
    domapis: function(nodeName, attributes, parent) {
        let elem = document.createElement(nodeName);
        for (let attr in attributes)
            elem.setAttribute(attr, attributes[attr]);
        document.querySelector(parent).appendChild(elem);
    }
});
```

**Listing 1: Load Wasm and expose JS and DOM APIs to it**

```
#[wasm_bindgen]
pub fn concat_strings(first: &mut String, second: &Vec<String>,
    prepend: bool) {
    for part in second.iter() {
        if prepend {
            first.insert_str(0, part);
        } else {
            first.push_str(part);
        }
    }
}
```

**Listing 2: Rust-based Wasm exporting a callable function**

The Wasm binary can call the *jsapis* and *domapis* functions in the JS environment to execute code or append elements to the DOM. Listing 2 shows a Rust program which once compiled to Wasm, will expose the function *concat_strings* to the hosting JS environment.

### 2.2 Workers and Worklets Contexts

They are independent threads meant to offload the computation burden on the UI thread to carry out specialized tasks, typically during the rendering process (e.g., audio, shared storage, or CSS paint worklets) [26], perform heavy computations for a single context (e.g., dedicated workers) or multiple contexts (e.g., shared workers) or persist in the background beyond the termination of other contexts (e.g., service workers). As shown in Listing 3, a context starts a dedicated worker, and listens for messages it sends, before sending a message to it.

```
let worker = new Worker("worker.js")
worker.onmessage = (event) => { /* ... */ }
worker.postMessage("...")
```

**Listing 3: Spawn and interact with a dedicated worker**

Indeed, worklets, workers, and browsing contexts execute in isolation, i.e., they do not have direct access to one another contexts via JS or Wasm. Table 2 lists an excerpt of APIs exposed to workers and worklets contexts. As one can read, worklets are heavily

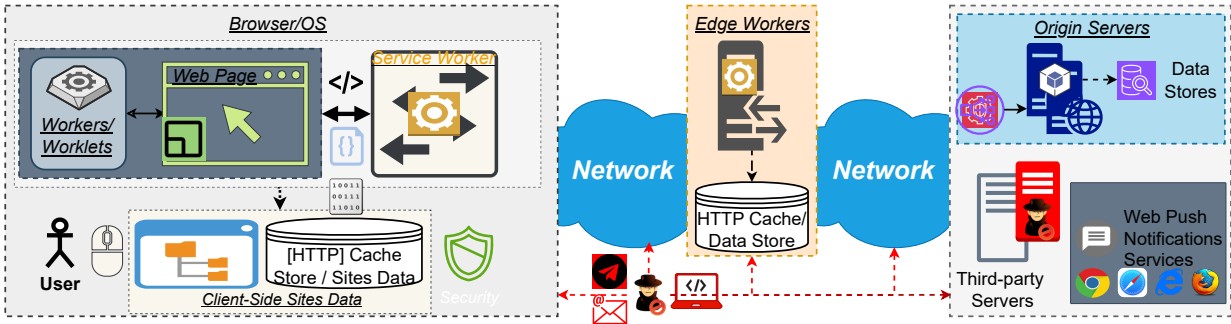

**Figure 2: Modern Web Architecture**

| API, Events, Features | Service Worker | Shared Worker | Dedicated Worker | Worklets |
|---|---|---|---|---|
| *DOM and BOM* | | | | |
| global object (this) | ✔ | ✔ | ✔ | ✗ |
| DOM (document) | ✗ | ✗ | ✗ | ✗ |
| Location object | ✔ | ✔ | ✔ | ✗ |
| *Additional JS code and Wasm* | | | | |
| Wasm APIs | ✔ | ✔ | ✔ | ✗ |
| importScripts | ✔ | ✔ | ✔ | ✗ |
| setInterval, setTimeout, Function, eval | ✔ | ✔ | ✔ | ✗ |
| *Additional Dedicated Workers* | | | | |
| new Worker | ✗ | ✔* | ✔ | ✗ |
| new SharedWorker | ✗ | ✗ | ✗ | ✗ |
| *JS Events* | | | | |
| message | ✔ | ✔ | ✔ | ✗ |
| connect | ✗ | ✔ | ✗ | ✗ |
| notificationclick, fetch, push, sync, install | ✔ | ✗ | ✗ | ✗ |
| *Connection APIs* | | | | |
| fetch | ✔ | ✔ | ✔ | ✗ |
| XMLHttpRequest | ✗ | ✔ | ✔ | ✗ |
| navigator.sendBeacon | ✗ | ✗ | ✗ | ✗ |
| *Storage Mechanisms* | | | | |
| CookieStore, IndexedDB, cache, file system | ✔ | ✔ | ✔ | ✗ |
| webStorage, document.cookie | ✗ | ✗ | ✗ | ✗ |
| *SOP restrictions workers locations* | | | | |
| SOP-restricted | ✔ | ✔ | ✗ | ✗ |
| CORS-restricted | N/A | N/A | ✔ | ✔ |
| Local URIs | ✗ | ✗* | ✔ | ✔ |
| *CSP directives that govern workers creations* | | | | |
| worker-src | ✔ | ✔ | ✔ | ✗ |
| script-src-elem | ✗ | ✗ | ✗ | ✔ |

**Table 2: Excerpt of Workers and Worklets Features**

restricted to the JS core [44]. The DOM is solely a browsing context privilege, which worklets and workers neither have nor can access others'. Hence, cross-context (post)messaging APIs can be relied on to exchange data. Cross-context shared memories can be accessed with the *SharedArrayBuffer* API [19]. Service workers are privileged workers supercharged with various events, including the lifecycle *install* or *activate* events fired when they first register or update, and many other functional events like *fetch* (for intercepting and responding to HTTP requests), the *push*, *notificationclick*, *notificationclose*, along with the *PushManager* and *Notification* APIs for managing web push notifications and messages sent via vendors services; the *sync* or *periodicsync* for background syncing, etc.

In addition to the eval-like functions, workers have access to the *importScripts* function dedicated to pulling down and executing additional (third-party) scripts. Notably, dedicated workers can spawn additional dedicated workers. In Firefox, shared workers can also spawn additional dedicated workers. Creating shared or service workers is only allowed from browsing contexts.

## 2.3 Client-Side Storages

With the notable exception of web storage (i.e., *localStorage* and *sessionStorage*), web workers have access to other storage mechanisms. In particular JS-accessible cookies can be manipulated via the asynchronous *CookieStore* API, counterpart to *document.cookie* in web pages. The Cache API is a programmatic storage, chiefly leveraged in service workers contexts, to store and retrieve HTTP requests and responses. Its benefits range from performance improvements to the ability to provide offline experience to users. The file system (FS) is a recent JS API that brings file management capabilities to websites, accessible to pages and workers. It has two instantiations. On the one hand, the origin private file system (OPFS) exposes a virtual file system whose data is persisted as other client-side storage in a location chosen by the browser, i.e., typically the browser profile folder. The file system access (FSA) API on the other hand, is an extension currently only supported by Chromium browsers, which gives direct read and write access to the user's physical file system, provided that the user has explicitly granted the permission, and the selected location is not blocklisted.

## 2.4 Edge Workers

In the traditional (or so simplistically abstracted) client-server Web architecture, responses are generated at the origin server side and consumed at the client side where the requests are issued. It is known that the Web architecture is more complex and comprises many relays and links that transport requests and response data. Among those, proxies as varied as the specialized services they provide. They can be transparent or passive, in the case their role is limited to simply relaying requests and responses without modification. Non-transparent or active proxies on the other hand may alter the passing exchanges when they are intercepted. Proxies controlled by the web author are referred to as reverse proxies, while those controlled by the client are forward proxies [18]. We further make the distinction *static* and *dynamic* (or programmatic) to tell

apart different types of edge proxies. In the former, we include most CDNs that retain the programmatic logic but offer the caching of static files to web authors via a configurable set of rules [40, 41]. In the latter, we include serverless applications and workers which gives developers the ability to define fine-grain caching and routing logic for the content on the edge. Notable examples are Cloudflare and Akamai workers, etc. [2, 5].

## 3 Cross-Site Scripting (XSS)

This section is dedicated to XSS, its threat model (Figure 2), and taxonomy (Figure 1). It ends with a summary of related work.

### 3.1 Threat Model

The typical *web attacker* [27] is a third-party entity that exploits weaknesses in the design, implementation, or deployment of a benign but buggy website to undertake unwanted actions that undermine the security and privacy of the user. In this category, we can include cross-site scripting (XSS), a common Web attack where an injection flaw is exploited to send an unsuspecting user a malicious code that will execute in their browser. The attacker is tech-savvy and can operate programs as diverse as network software, and HTTP servers, to host various types of content and code, e.g., JavaScript files. It is assumed that the attacker can leverage social engineering tactics [23] to deliver maliciously crafted links to users via means as diverse as emails, social networks, messaging apps, and attacker-controlled websites. Users are then incentivized, teased, or tricked into navigating the malicious links in browsers where they are or are potentially interacting with websites entrusted with sensitive information and data.

### 3.2 Current Taxonomy

Figure 1 presents the current XSS taxonomy adopted by the community to frame the different types of XSS attacks, i.e., reflected (non-persistent) or stored (persistent) server or client XSS [34, 55]. DOM-based XSS [7] is categorized as a subset of client XSS, and persistence is understood as the likelihood of success of an exploit. The server or client defines the vulnerable component responsible for the *injection* of the malicious payload in the website. In server-reflected XSS –typically targeting a single user at a time and considered the simplest to find and defend against– a request URL argument is for instance included in the generated response. An instance of server-stored XSS –considered the most dangerous as it can persist and affect multiple users indefinitely– would involve the storage of attacker input (e.g., HTML form data) in a SQL database, and its later inclusion unsanitized in HTML responses sent to multiple users. Client XSS are all the other categories where the vulnerability resides exclusively in the client-side code. A non-persistence example would be the unsafe usage of attacker-controllable sources like the URL (e.g., the *location* object) into sensitive sinks like *eval, document.write*, etc. A persistent variant would be one leveraging the cookie or localStorage for instance [48].

### 3.3 Related Work

By far, web pages are the contexts which have gathered most attention from the research community as well as Web authors, browsers

and other defenses vendors [22]. Regarding the client-side, non-persistent DOM-based have been extensively studied [37, 39]. Comparivetely stored XSS has gained limited consideration from the research community. Steffens et al. [47] demonstrated instances of it with the usage of local storage and cookies. The community has also demonstrated other XSS variants exploiting JavaScript gadgets [36], the mutation of the DOM [33] or vulnerable events handlers, in particular the message event [48]. These injections do not trigger code execution *immediately*. Instead, code executions occur due to quirks and specific behaviors imputable to transformation brought to the HTML or DOM by browsers or web frameworks. These variants have somehow demonstrated the limitations of the XSS taxonomy at properly framing complex types of attacks, referred to as code-reuse or mutation XSS. In the current XSS taxonomy, these attacks can be categorized as either client-side or server-side based on how the JS gadgets or the mutable DOM elements are injected in web pages. Server XSS can be discoverd for instancee by searching for the presence of URL parameters in HTTP responses. Stored instances are more tricky to unveil without access to the logic and code of the server [21].

Research on workers XSS is more seldom. It was with the advent of service workers that the first studies have been done. Chinprutthiwong et al. found non-persistent [30] and persistent XSS [31] uses of the *importScripts* sink due to the import of code from two sources which can be controlled by a malicious script in web pages: the servcie worker URL and the IndexedDB storage. In the case of the service worker URL, the authors found a handful websites who could be infected by a traditional web attacker via the URL of the web pages registering the service worker. For other cases, they assume a strong attacker who exploited an initial vulnerability to execute a code in a same-origin context in order to further affect the service worker's logic. Squarcina et al. [45] also assumed a strong attacker and demonstrated that the Cache API is a vector that can be leveraged to store malicious codes in HTTP responses, and have them served by service workers and executed in other contexts like web pages, shared and dedicated workers and worklets. From a server-side perspective, Watanabe et al. [50] demonstrated that in shared hosting settings like web archives, proxies or translators where initially cross-origin websites are gathered under a single (same-) origin, attackers could host malicious content (web pages) and service workers, and once they have users visit pages under their control, they register the malicious service workers. We note the work of Subramani et al. [49], discussing the ability of Firefox extensions to tamper with HTTP responses, or third party libraries deliberately included by web authors, to hijack service workers and execute malicious code.

## 4 Motivations

Considering the variety of stakeholders and technologies, the Web attack surface runs from the clients where requests are issued, to the servers where responses are generated, as well as the related components including user agents (i.e., browsers) and new features they constantly propose, users themselves and their security-awareness, their devices and underlying operating systems security, and any

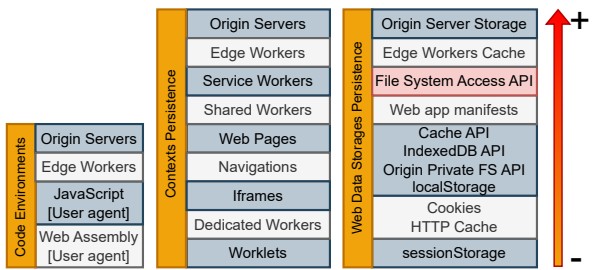

**Figure 3: Precedence and persistence of executions environments, contexts and storage mechanisms**

intermediate middleware that can act on the HTTP communications and data exchanged over the network. Figure 2 shows the position of the main components in the Web architecture.

## 4.1 Code Execution Environments and Contexts

As one knows, there is more than browsing contexts in user agents, and malicious code execution is a concern not only to JavaScript. As an additional environment, Wasm can execute malicious code. Likewise, web workers and worklets are contexts solely dedicated to code executions and can therefore be targets of malicious code injection and execution. In the code snippet of Listing 1, assuming that the URL */wasm.wasm* of the Wasm binary comes from an attacker-controllable source like the *location* object, one would achieve code execution, not in the JS environment per se, but in Wasm. Similar examples can be devised for workers and worklets APIs (See Table 1). Indeed, it is the current design of the Web that JavaScript in web pages serves as a host (it precedes) for loading Wasm, and the initial workers and worklets. Figure 3 presents the precedence and persistence of these components we have discussed so far and Section 5.1 discusses their importance in describing code injection attacks.

## 4.2 Cross-Browser Permanent [FS] Storage

Most client-side storage mechanisms are subject to security restrictions. Cookies scope can be at most all websites sharing a common top domain (eTLD+1) [16], but can also be restricted to a single domain (or origin), or down to a single path (or resource). Other mechanisms like web storage (localStorage, sessionStorage), IndexedDB, the cache API, or the origin private file system instantiation of the file system API, are restricted by the SOP. That is with the exception of the FSA API. Within the same browser, the SOP does not apply, and websites are not prevented from concurrently reading and writing the same locations. Additionally, across browsers, no restrictions are applied: different websites from different browsers can access the same locations on the user's file system. This is the first native and permanent cross-browser JS-accessible client-side storage introduced by browsers. This FSA API has no precedence on the Web and changes the perspective we have on the persistence of client-side storage: content on the file system can outlive all other mechanisms, including browsers themselves, their updates, reinstallations, and removals from users' devices.

Figure 3 presents the suggested precedence of different storages. In Section 5.2 we redefine persistence as a continuum, relating to not only storages but code execution environments and contexts as well.

## 4.3 Dynamic HTML, JS, and Wasm Responses

Infecting a web app with server XSS, i.e., having it generate dynamic responses based on attacker inputs, has been so far only considered from the perspective of web pages, i.e., HTML/DOM injections. We argue that DOM scripts, workers, and Wasm sources can equally be dynamically generated based on attacker-controllable inputs. This can be done at the origin server but also other HTTP endpoints like edge or service workers. We are only aware of prior attempts at studying dynamic DOM scripts. Unfortunately, they are either limited to JSONP endpoints [51] or are unrelated to code execution [38].

## 4.4 HTTP Endpoints Compromises

HTTP endpoints are so far seen as vulnerable closed-boxes, which the attacker probes with maliciously crafted inputs (sources) to have *dynamically generated outputs* which include code to be executed in the victim user's browser. First, if the role of the HTTP endpoint is naturally fulfilled by origin servers, edge proxies on the network and service workers in browsers now also share this capability. Moreover, we argue that the closed-box vision is incomplete. In fact, with the rise of supply chain attacks, and the reliance on third-party frameworks to implement the logic of critical components including service, edge workers, and the origin server itself, or the deployment of components on potentially adversarial settings like shared hosting on the cloud, the delegation of services to third parties in content management systems (CMS) like WordPress plugins [11, 14, 24, 25, 35], the likelihood of successful HTTP endpoints (and hence resources) compromises are high. Should a compromise arise, attackers gain scripting capabilities and can respond to HTTP requests with malicious resources. We note that the community has considered the eventuality of these attacks, and pushed for mechanisms like SRI based on checks on the integrity of resources hashes [20]. Unfortunately, this is only limited to DOM scripts and stylesheets, overlooking Wasm, workers, or worklets. Finally, even without scripting capabilities, the attacker could *statically* host (upload) (malicious) content (e.g., HTML, Wasm, JS) on HTTP endpoints by abusing a legitimate service or by exploiting a vulnerability e.g., a FTP (file transfer protocol) server hosted on the same machine or a path traversal flaw of the endpoint [12, 15, 50]. Specifically, if the uploaded content is a program (e.g., PHP or JS) interpreted as such by the vulnerable endpoint, the attacker further gains scripting capabilities and can choose the code to be injected in client-side execution contexts. In Section 6, we demonstrate an instance of this scenario with the *Service-Worker-Allowed* header.

## 4.5 Web Cache Collision

We recall that performance and potentially greddy caching are part of the features service or edge workers provide. First of all, we stress that it is an intended behavior that they do not have to adhere to HTTP caching headers served by origin servers. Hence, they can cache a resource indefinitely even though its HTTP cache headers

specify otherwise, e.g., *cache-control: no-store*. Furthermore, service workers can be configured to collide different requests by ignoring URL search parameters, the request method, or the *Vary* header [3]. We refer to this behavior as *cache collision*. Listing 4 shows a code snippet of a service worker effectively performing cache collision.

```
self.addEventListener("fetch", event => {
    event.respondWith(caches.match(event.request, {
        ignoreSearch: true, ignoreMethod: true, ignoreVary: true
    }));
})
```

**Listing 4: Cache collision during cache search**

Assuming that this code is deployed on *example.com*, Listing 5 shows different URLs which collide into a single one.

```
https://example.com/?name=attacker<script>alert(1)</script>
https://example.com/?name=johndoe
https://example.com/
```

**Listing 5: Cache collision treats different URLs as a single one, by ignoring the search parameters, request method, etc.**

The search parameters are ignored, and no matter whether the resources are accessed with a *post* or *get* method for instance, if a response to one of the URLs is cached, it will be served. Overall, in the presence of a cache collision flaw, either imputable to the Web author, or most certainly introduced by a third-party library, a reflected XSS attack performed by the attacker against an origin server, could be persisted on the edge, and automatically served to multiple users.

## 4.6 Web Push Messages Hijacking

Workers and web pages execute in isolation and are extensively event-driven. In particular, web apps can subscribe users to web push notifications and register service workers to receive and process push messages in the background even when the user navigates away from the app, and show notifications to engage them back with the web app. With permission granted to a website, an attacker can *resubscribe* the user with credentials that they control, obtain a subscription endpoint, and leverage it to regularly send messages to the service worker. If the latter has a vulnerable push message handler, the attacker can send malicious messages to the underlying browser web push notification service (See Figure 2), which will be delivered to the user's service worker and executed. This can be done whenever the attacker chooses to.

## 5 Revisiting the XSS Taxonomy

We note that there is always strong resistance to questioning a well-established area of research, like XSS. Nonetheless, we think that in light of all the discussion that precedes, we can make this first step forward. We hope to have shown that the Web and XSS have gotten complex enough to demand at least a revisit of the current taxonomy. Due to page limits, we refer reader to the Appendix on how we express different attack scenarios with our taxonomy in Figure 5.

**Yet another taxonomy?** Foremost, our goal is not to get the community out of the current XSS matrix, and propose another rigid one, because it will certainly show its limitations as the Web evolves. Rather, our proposal is to allow for a better description of code injection attacks: the components (i.e., environments, contexts, storages) and actors (i.e., victim user or adversary) involved as well as the recurrences in the delivery of the malicious payload (infection), and the resulting code execution (affection) triggering processes. While accommodating the prior taxonomy, our proposal allows better coverage when assessing the extent and susceptibility of a website to (code injection) attacks. Figure 3 shows these main components, as well as their precedence in terms of persistence and hosting.

**Attacker and Capabilities** Figure 2 presents the position and capabilities of the attacker in this ecosystem. We consider modern and up-to-date user agents (e.g., Web browsers), correctly implementing Web standards, and enforcing state-of-the-art security principles like the SOP. Then, we consider a traditional web attacker [27] with the following clarifications. From a web author's perspective, we only consider those HTTP endpoints, nodes, or components under their control, and which are potential vectors to code injection attacks. Notably, we include benign-but-buggy, rogue, compromised, or malicious third-party programs that the Web author has deliberately linked to in their app, e.g., pages, service, or edge workers. In line with many prior work, we also include a strong attacker, i.e., an attacker who has already gained code execution in the website, as this can serve as springboard to mounting more sophisticated attacks, in particular persistent ones or those which target worker contexts [30, 31, 45, 47]. Furthermore, it is common to assume that websites are susceptible to code injection attacks as setups for various code injection studies [33, 36]. However, we exclude network attackers, as they can be trivially thwarted by deploying secure communications protocols, e.g., HTTPS. In the same vein, we exclude from our threat model, browser extensions, plugins, and potentially ill-intentionned forward proxies and any intermediate component deployed by users or under their control (e.g., an HTTPS Mitmproxy [32]).

**A problem of branding?** JavaScript is a scripting language, which is reflected in XSS naming. But Wasm is compiled, not *scripting*. Moreover, the community agrees that the cross-site terminology is misleading [6]. Because they always carry malicious purposes, we think that the term *malicious code injection and execution* expresses more the essence of the attacks we are concerned with. Hence, we put forth the OWASP definition of *XSS as a code execution flaw [34] where attacker-controlled data is sourced into a sensitive sink API/function where it is executed as code in a vulnerable website rendered in a victim user browser*. We use the terms *infection* and *affection* as synonyms for *injection* and *execution*. As for *code*, it is a special type of the general term *content* that covers other non-code scenarios. We note that injection can occur without execution (if defenses are in place), and are independently or altogether worth assessing.

## 5.1 Components, Recurrence, and Actors

In the current taxonomy, the idea of server-/ or client-side indicates whether a server (i.e., origin server) or client component (i.e., web page) is involved in the injection process. As for reflected or stored, they have to do with the likelihood of a successful exploit (execution).

**Specifying the component** We do not argue against the employment of the server or client terminology, but rather to not reduce them respectively to the origin server or the DOM (web pages). Instead, we argue for always specifying the component (e.g., origin

server, edge worker, web page, or service worker) to help clarify the description of a code injection attack. In the current taxonomy, a server reflected XSS is presented as such without further details, while one knows that from the perspective of a user, the HTTP endpoint responsible for such injection could as well the origin server, an edge worker, or even a service worker laying in their browser. Additionally, multiple components could be involved in a code injection flaw, e.g., a payload reflected at an origin server, but stored on the edge.

**Injection and execution properties**. We introduce two additional important properties. The *recurrence* expresses the number of users infected and affected by a malicious payload delivery and its execution. The *actor* is the entity (user or attacker) that ultimately triggers the injection or execution. We note that the attacker is always the adversary, and the user the victim. To infect users for instance, it is the users themselves who ultimately trigger reflected XSS (e.g., by following a malicious page link), while the attacker triggers server-stored XSS (e.g., by sending a malicious form directly to the origin server). As for the affection (exploit), in most cases, it is triggered by the user (e.g., by visiting an infected page). As we have shown in Section 4.6, a vulnerable push message handler execution can be directly triggered by the attacker by sending the payload to the user's browser via a vendor's push service.

## 5.2 Persistence as a Continuum

So far, persistence has been understood as to whether a (persistent) storage is leveraged to store the malicious code. This vision is incomplete. We note indeed that environments, contexts, and storages can all be more or less persistent, e.g., service workers. Each component can then extend its persistence to the overall vulnerability. Moreover, not all components can be placed on the same level in terms of persistence, e.g., a sessionStorage which expires after a page closes, compared to an indexedDB that persists between different browsing sessions, or the file system which we dubbed permanent because it is more important than the persistence of other client storages. Figure 3 hints at the suggested non-strict ordering of persistence and precedence of environments, contexts, and storage. Therefore, the idea of persistence should be considered a continuum, rather than a binary marker persistent vs. non-persistent. Moreover, it should be expressed relatively to a particular component. With think that specifying the components involved in an attack (as discussed in Section 5.1) will help capture its persistent nature, as the attack can be dubbed persistent when its components are. We refer the reader to the Appendix on how we express different attack scenarios with our taxonomy in Figure 5

## 6 Service-Worker-Allowed Header Issues

We demonstrate a security issue introduced by service workers, with the *Service-Worker-Allowed* header. We take the example of a web server that hosts multiple user profiles with the ability for the members to use server-side programming languages (e.g., PHP) to generate their content. Figure 4 summarizes the attack setup and exploit. As shown, the attacker first tricks the user into visiting their profile, i.e., hosted under *~/attacker/* on the web server. In response, a malicious service worker included in the attacker's profile page is automatically registered in the background. However,

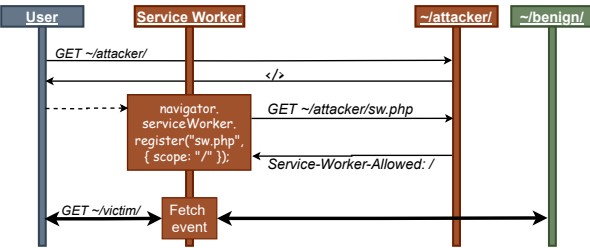

**Figure 4: Service Worker Allowed Header**

because the attacker service worker is located in an inner scope (i.e., *~/attacker/*), they leverage the server-side scripting capability to add the *Service-Worker-Allowed* header to the service worker response. By doing so, the malicious service worker gains control of the entire origin (i.e., the / scope) in spite of its being located under an inner scope. By setting up the *fetch* event, the attacker is able to record and/or tamper with all the data exchanged by the user with the web server. This could potentially include credentials used to log into other profiles on the website, and potentially even the site's author if they operate an admin profile.

**Attack relevance** This kind of shared hosting website is generally deployed at universities and research institutes, where an admin setups an Apache web server on a Linux machine, with PHP as the programming language for instance. Then, by enabling the *userdir* module, the admin grants each member (e.g., a faculty) credentials on the web server, to create and manage a subfolder of the website. We have mounted this attack in controlled environments, on different Linux systems, including Debian, Ubuntu, Fedora, etc. It is really as simple as we have just shown, as it is also simple to setup an Apache web server with PHP, and enable the *userdir* module. We note that the potential security problems in this kind of shared hosting environment have been known for years because the users share the same origin. For instance, an attacker (who controls a part of the website), can trick users into visiting a same-origin page under their control, where they can read/write authentication cookies or issue requests on behalf of the user. To prevent this kind of issue, cookies can be scoped to specific paths, e.g., *~/username*, or disabled altogether. Nonetheless, we observed that this kind of website is still used today, in particular in (reknown) universities and research institutes. That notwithstanding, with this attack, we demonstrate that there is a completed new exploit opportunity that service workers have brought: the ability for an attacker to take control of the whole origin, i.e., the whole / scope, even though the attacker can only host content under an inner scope */~attacker*.

## 7 Defenses against XSS

To successfully mitigate content injection attacks, stakeholders usually advise the deployment of various lines of defenses [6, 36].

### 7.1 Input Sanitization and Output Encoding

With input sanitization, an application ensures that the data it is processing is of an expected, valid, and safe type. Then, when such input is to be included in a response, the application should apply context-aware encoding to ensure that the output would not

trigger undesirable side-effects when rendered by the destination context (e.g., execution in browsers). For instance, HTML tags like *<script>* must be filtered out and one has to ensure that strings are treated as such and not interpreted as code if they happen to be part of HTML elements event handlers (e.g., *onload="..."*). A web application firewall for instance can fulfill many filtering tasks based on rules to be matched against requests. From the client side, one has to ensure that APIs that manipulate or augment the DOM (e.g., *document.write*, *HTMLElement.innerHTML*) are not passed arbitrary attacker-sourced parameters or are purified before hand [8].

The community is well aware of JS contexts in HTML responses. [6]. Expectedly, workers, worklets, and Wasm are overlooked. For these contexts, there are no HTML tags (e.g., *<wasm>*) or code execution attributes (*oncompiled="..."*) to indicate a code execution context. Furthermore, in addition to eval-like functions, the *importScripts* function is a specific code execution sink in worker contexts. Hence, defenses solely deployed at the origin server side to filter out or block HTML elements, or a set of dangerous functions (e.g., at a JSONP endpoint handler), will miss the discussed instances, and malicious activities in service or edge workers.

| Directive Names | Supported Source Values |
|---|---|
| `script-src`, `default-src` | hosts, schemes, *'none'*, *'self'*, *'unsafe-eval'*, *'wasm-unsafe-eval'*, *'report-sample'* |
| `sandbox` | allow-scripts, allow-same-origin |
| `worker-src`, `child-src` `connect-src` `base-uri` | hosts, schemes, *'none'*, *'self'* |
| `report-uri`, `report-to` | *groupname*, ⟨*uri*⟩ |
| `require-trusted-types-for` | *script* |
| `trusted-types` | foo, bar, ... |

**Table 3: CSP expressiveness in workers contexts**

## 7.2 Content Security Policy (CSP)

CSP has imposed itself as one of the defense-in-depth and widely scrutinized mechanism for mitigating XSS in browsers [28, 29, 42, 43, 46, 51–54]. It has approximately 30 directives for expressing restrictions against the fine-grained types of content that web pages embed (e.g., *img-src* for images, *frame-src* for iframes, etc.) [52, 53]. Workers on the other hand operate a handful of types of resources, and a tiny part of CSP is relevant in these contexts, let alone the expressiveness that is honored by browsers. Table 3 presents the CSP features [52], and how they apply to different workers.

**Code Execution** The `worker-src` directive which defaults to `child-src` and then to `default-src`, is relevant in dedicated and shared workers only, as those can spawn additional workers (See Section 2.2). As for the `script-src` directive, which defaults to `default-src` when missing, accommodates JS code execution restrictions, for instance, trusted hosts and schemes, including local schemes (e.g., *data:*, *blob*). Notably, features like hashes or nonces are not supported. The *wasm-unsafe-eval* keyword, which defaults to *unsafe-eval* when missing, allows specifically the execution of Wasm. For worklets, as they can not load additional code, only the *report-sample* keyword is relevant. As for the `sandbox` directive, only the two tokens that control the SOP or code execution are enforced. At this point, it is important to make the following observations. Let's consider the two policies in Listing 6.

```
content-security-policy: default-src 'none';
content-security-policy: sandbox 'allow-same-origin';
```

**Listing 6: Controlling code execution with CSP and**

In browsing contexts, these policies have the same effect of disallowing code execution altogether. In worker contexts, however, only the sandboxing disallows code execution altogether. The first policy only disallows additional code that the worker may attempt to load, via APIs like *importScripts* and eval-like functions. But the initial worker code (e.g., from an origin server) will execute.

**Connections and service workers specifics**: Workers can connect to remote servers, and hence the `connect-src` directive is relevant for constraining the connection endpoints. This includes HTTP, web sockets, and server-sent events. As shown in Table 2, XHR is not supported in service workers contexts, because of its blocking capability, and no worker contexts can send beacons. Importantly, we note that requests initiated by other client contexts (i.e., web pages, dedicated and shared workers under their scope or control) go through the *fetch* event handler of service workers (See Listing 4). No matter their initial types (e.g., images, stylesheets, videos, same-origin iframes, forms, web sockets) – which are indicated by the destination of the request [17] object –, all requests can be passed to the *fetch* API to download the related responses from backend servers. That is how it happens that only the `connect-src` directive is relevant in (service) workers contexts. Finally, we observe that requests passed as-is to the the fetch API within a *fetch* event handler, will have their fetch metadata headers [9] preserved when received at the backend HTTP endpoint. However, when service workers modify the requests before passing them to the *fetch* function, their fetch metadata headers are altered, and they are sent as *fetch* requests, no matter their original types.

## 8 Conclusion

To the best of our knowledge, this is the first study to question the traditional understanding of XSS and its taxonomy, which fails to properly account for additional code injection vectors and components in an ever-complex Web architecture. We propose a framework where one can precisely define the components –i.e., environments like JS or Wasm, contexts like (service or edge) workers, and the storage mechanisms (e.g., cookies or the file system)– as well as the actors (user or attacker) and their recurrence in the infection and affection process of code injection attacks. Overall, as XSS in web pages has been endorsed widely and quickly by the community. This has led to extensive studies and incrementally improved defenses. We hope that the awareness we raise about the larger attack surface will lead to more defensive and effective solutions for other contexts and environments. As the Web evolves, caution and a thorough analysis of the extent of code injection attacks should be undertaken first, when vendors add support for features that deeply modify the Web attack surface.

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

| | Stored Origin | Reflected Origin | Location DOM | Steffens et al.[47] | Steffens and Sock [48] | Chinprutthiwong et al. [30, 31] | Squarcina et al. [45] | Watanabe et al.[50] | FSA API | SWA Header | Cache Collision | Web Push Messages |
|---|---|---|---|---|---|---|---|---|---|---|---|---|
| **Infection: Actor ↦ Recurrence** | A↦1+ | U↦1 | U↦1 | U/A↦1 | U↦1 | U/A↦1 | A↦1 | A↦1+ | A↦1 | A↦1+ | A/U↦1+ | A/U↦1 |
| **Affection: Actor ↦ Recurrence** | U↦1+ | U↦1 | U↦1 | U↦1+ | U↦1 | U↦1+ | U↦1+ | U↦1+ | U↦1+ | U↦1+ | U↦1+ | A↦1+ |
| *Vulnerable Code (Infection)* | | | | | | | | | | | | |
| **Origin Server Code** | ✔ | ✔ | | | | | | ✔ | | ✔ | ✔ | |
| **Edge Worker Code** | | | | | | | | | | | ✔ | |
| **JS [User agent/Client]** | | | ✔ | ✔ | ✔ | ✔ | ✔ | | ✔ | | ✔ | ✔ |
| **Wasm [User agent/Client]** | | | | | | | | | | | | |
| *(Malicious) Code Execution Contexts and Environments (JS/Wasm)* | | | | | | | | | | | | |
| **Worklets** | | | | | | | | | | | JS | |
| **Dedicated Worker** | | | | | | | | | JS | | JS | |
| **Iframes** | | | | JS | | | JS | | | | JS | |
| **Navigations** | | | | JS | | | | | JS | | JS | |
| **Web Pages** | JS | JS | JS | JS | JS | | JS | | JS | | JS | |
| **Shared Worker** | | | | | | | | | JS | | JS | |
| **Service Worker** | | | | | | JS | JS | JS | JS | JS | JS | JS |
| *Storage Mechanisms* | | | | | | | | | | | | |
| **Origin server Storage** | ✔ | | | | | | | ✔ | | ✔ | | |
| **Edge worker Cache** | | | | | | | | | | | ✔ | |
| **File system access API** | | | | | | | | | ✔ | | | |
| **Web app manifests** | | | | | | | | | | | | |
| **Cache API** | | | | | | | ✔ | | | | ✔ | |
| **IndexedDB API** | | | | | | ✔ | | | | | | |
| **Origin private file system** | | | | | | | | | | | | |
| **localStorage** | | | | ✔ | | | | | | | | |
| **HTTP Cache** | | | | | | | | | | | | |
| **HTTP Cookies** | | | | ✔ | | | | | | | | |
| **sessionStorage** | | | | | | | | | | | | |
| *Sources (URL/Storage, etc.) / Sinks (HTML/JS/Wasm)* | | | | | | | | | | | | |
| **Source** | URL (Body) | URL | URL | URL/ Storage | Message Event | URL/ Storage | Storage | Server File | Storage | Server Program | URL | Push Event |
| **Sink** | HTML | HTML | HTML/JS | HTML/JS | HTML/JS | importScripts [JS] | HTML/JS | JS | JS | JS | HTML/JS | JS (*eval*) |

Figure 5: Demonstration of the usage of the taxonomy to frame prior and current attacks discussed in this work. Actors (A) is the attacker and (U) is the user. The recurrences 1/1+ means exactly 1 or more than 1. We also included some common sources and sinks to help the reading, though those are out of the scope of the taxonomy. For infection, if the attacker can directly infect a web server (e.g., the origin server), they cannot directly infect the user's browser without the user involved, unless they already gain code execution in one of the website's contexts. This is the attacker model of most work on web workers, and one of the attacker models of Steffens et al. for persistent client XSS [30, 31, 45, 47]. Likewise, the user is the one who ultimately triggers code execution in their browser in most cases, with the notable exception of a malicious web push message which is an attacker's privilege. In terms of recurrence, when an origin server or an edge worker is involved, infection typically reaches many users, otherwise only a single user is infected. Obtaining multiple occurrences of code execution is achieved when there is persistence, provided either by a storage mechanism, or a persistent context like a service worker. We note that there is no attack scenario or prior work that involved Wasm in either the infection or affection processes, as all the cases are related to vulnerable or malicious JavaScript. Expectedly, web pages have been extensively considered for code execution, as well as service workers. The analysis of other contexts are interesting directions for future work. Unless otherwise specified, we also include in the web pages, iframes, and other top-level navigations initiated programmatically from a browsing context. As one can observe, many storage mechanisms are overlooked in the literature and are therefore interesting directions for future work. Events may also deserve deeper scrutiny. Finally, Wasm is overlooked as a potential code execution sink by itself, as well as dynamic JavaScript.

## Browsing Contexts

These are threads where takes place the rendering of root web pages (e.g., in top-level browser tabs), as well as nested documents (e.g., iframes). Navigating different pages can be statically initiated by users entering URLs in user agent (e.g., web browsers) interfaces, clicking links, or submitting forms from other pages. Such navigations can also be dynamically triggered by JavaScript APIs like *window.open*, *clients.openWindow*, or *location* object, etc. The document is presented for interaction with the user in an interface (UI) −referred to as the main UI thread−, and for manipulation to JS in the DOM API. Among the numerous ways JS can be included in web pages is the use of the *<script>* tag to provide inline code directly or reference a remote resource (with the *src* attribute).