# OpenReview forum: "MatriXSSed: A New Taxonomy for XSS in the Modern Web"
_ACM.org/TheWebConf/2025/Conference — WWW 2025 Poster_

### Official Review · Reviewer_naxi · 2024-11-24

**Novelty:** 4
**Technical Quality:** 5

**Review:**

This paper revisits the taxonomy of XSS attacks to better integrate modern web technologies, including Wasm, service workers, edge workers, and File System Access API. The authors propose a descriptive framework that accounts for these attack vectors and demonstrate its utility with examples.

Pros:
1. The paper identifies limitations in the existing OWASP XSS taxonomy, particularly its inability to describe attacks involving new components like Wasm or service workers.
2. The proposed taxonomy introduces key dimensions of XSS attacks, such as persistence, recurrence, and actors.
3. The insights into service worker vulnerabilities, especially in shared hosting environments, are impactful.
4. The examples provided, including the Service-Worker-Allowed (SWA) header abuse, highlight concrete security risks.

Cons:
1. Some discussions, such as the persistence continuum and component-specific vulnerabilities, are conceptually dense and could benefit from further simplification or visual aids.
2. The treatment of Wasm remains more theoretical, with limited demonstration of its use as an XSS attack vector. The practical adoption and feasibility of the proposed taxonomy by developers, researchers, and security practitioners are not sufficiently addressed.
3. The proposed taxonomy does not address how existing defense tools can adapt to these changes, which could limit its practical impact.
4. The paper's writing assumes a high level of familiarity with web security concepts, limiting its accessibility to a broader audience. Additionally, the lack of explicitly stated contributions in the introduction, unclear transitions, and uneven section organization hinder the clarity and coherence of the work.

**Questions:**

1. The paper critiques the current taxonomy as incomplete for modern web architectures. Could you provide specific metrics or evidence to support the claim that your taxonomy offers better coverage or detection rates for modern XSS attacks?
2. Your taxonomy includes various dimensions like actors, contexts, and storages. Have you tested its comprehensiveness against recent, high-profile XSS cases? If so, how does it compare to the traditional taxonomy in those scenarios?
3. The paper discusses persistence as a "continuum" rather than a binary property. Could you elaborate on how this continuum is defined quantitatively or qualitatively, and how it can be applied in practice?
4. The paper emphasizes Wasm as a new vector but without examples. Could you include a deeper exploration of Wasm-related XSS risks, supported by experiments or case studies? Could you offer practical guidance for developers to mitigate risks highlighted in the paper.
5. The distinction between "infection" and "affection" is conceptually interesting but potentially confusing. Could you provide additional examples or definitions to clarify them?
6. How would you envision the adoption of your taxonomy in current static or dynamic analysis tools? Could you provide a roadmap for integrating the taxonomy into existing security practices?

**Reviewer Confidence:**

3: The reviewer is confident but not certain that the evaluation is correct

**Scope:**

4: The work is relevant to the Web and to the track, and is of broad interest to the community

---

### Official Review · Reviewer_xbXc · 2024-12-01

**Novelty:** 4
**Technical Quality:** 4

**Review:**

## summary
Cross-site scripting (XSS) remains one of the most prevalent attacks on the web. However, the existing XSS taxonomy is insufficient, as the web has evolved extensively and the attack surface has expanded significantly with the introduction of numerous new technologies and features. Therefore, the authors question the traditional understanding of XSS and its taxonomy and propose a framework to define the components.
## Comment
First of all, the authors challenged the traditional taxonomy of XSS and proposed a step forward in revising the taxonomy. However, I would still like to make a few suggestions. Hope my opinions/suggestions will help improve your paper.
- **Redundant Sections.** In Section 2, the authors discussed the background of web assembly (Wasm), worklets and workers, client-side storage, and edge workers. However, Section 3 discussed the threat model, current taxonomy, and related work of cross-site scripting (XSS). These topics are closely related and could be combined into a single, more concise section for better clarity and cohesion. Additionally, the main content begins on page 6, despite the document being only 8 pages long.
- **Lack of Novelty** The paper focused on the taxonomy and understanding of XSS, proposing a framework to define its components and offering some defense recommendations. However, it lacked novelty and empirical support. There are no specific statistics or experimental results to substantiate their claims, relying solely on theoretical arguments. While it served as a supplement to the existing taxonomy, it fell short of presenting sufficient contributions to warrant a standalone publication.
- **Minor Issues.**
  - In Section 1, there is a typo "(Sections 3 and refsec:defenses)".
  - In Figure 5, the format of "JS" is not consistent.

**Questions:**

## Questions
- Can you make the paper more concise?
- How does your proposed framework or taxonomy improve the understanding of XSS in the context of modern web technologies? Can you provide additional evidence or empirical results to validate your taxonomy and advice? How does your taxonomy quantitatively improve on traditional XSS taxonomy?

**Reviewer Confidence:**

2: The reviewer is willing to defend the evaluation, but it is likely that the reviewer did not understand parts of the paper

**Scope:**

4: The work is relevant to the Web and to the track, and is of broad interest to the community

---

### Official Review · Reviewer_56Mz · 2024-12-02

**Novelty:** 6
**Technical Quality:** 4

**Review:**

The paper introduces a novel taxonomy for XSS attacks on the web, which takes into account the recent evolution and technology support for attack surfaces that were not (or only rarely) present when the initial taxonomy was formulated. For instance, WebAssembly, background services (push notifications) and workers/worklets.

The proposal aims at removing the rigid binary characterization of the XSS attacks and migrate towards a spectrum and a set of aspects that allow for more flexibility to specify the different actors, components, environments, storages, persistence and recurrence. As two examples of attacks that would be difficult to classify according to the existing OWASP matrix, the authors present a web push messages hijacking scenario and a service-worker-allowed header one.

The expanded support of web assembly and the other modern web features discussed in this paper are sufficiently described and they lend support to the thesis of this paper. While I am not a web security expert, I do appreciate the clarity of the exposition and the examples that are provided. I do believe that the main contribution of this paper, i.e., the new taxonomy and its represented usage in Figure 5, should be part of the main text as it gives a concrete example of how it would be used to classify existing attacks.

I believe that it would be interesting to have a deeper analysis of how new taxonomies have emerged in other, possibly related fields, and compare the process that the authors adopted in order to come up with this one (e.g., https://ieeexplore.ieee.org/document/9464576). The authors referenced existing literature that presented attacks that would be challenging to classify according to the existing OWASP matrix but did not evaluate their proposed new taxonomy in sufficient detail. I would suggest to try to abstract some of the specific works and try to come up with a higher-level categorization of some of these attacks, which could then support the proposed new taxonomy.

**Questions:**

* A more thorough comparison with how other taxonomies in similar domains have been proposed
* A possibly more structured (e.g, hierarchical if it makes sense) way to reason about the taxonomy

**Reviewer Confidence:**

1: The reviewer's evaluation is an educated guess

**Scope:**

4: The work is relevant to the Web and to the track, and is of broad interest to the community

---

### Official Review · Reviewer_LHXz · 2024-12-03

**Novelty:** 4
**Technical Quality:** 4

**Review:**

**Summary**

This paper discusses the limitations of the traditional taxonomy of XSS attacks according to the well-known matrix by OWASP, which categorizes the attacks either as server-side or client-side, and as reflected (non-persistent) or stored (persistent). The paper argues that this taxonomy is insufficient in capturing the complexities of the modern Web, considering the plethora of features and technologies that have emerged, which introduce new attack vectors and can be utilized for launching and executing such attacks. As such, the authors propose a revised taxonomy for XSS attacks, now referred to as "code infection and affection" attacks, that specify the different components (e.g., web technology like Service Workers) and actors (e.g., user or attacker) involved in the attack, as well as the recurrence of exploits. Furthermore, they introduce the idea of persistence as a continuum rather than a binary marker, since the various code execution environments, contexts, and storage mechanisms can all have different levels of persistence. By revisiting the XSS taxonomy, the authors aim to provide a more descriptive framework that adapts to the evolving web architecture and highlights the need for updated, context-aware security measures.

**Detailed comments**

* The paper is well written in general and is easy to follow and understand. It presents a plethora of attacks that are now feasible due to the introduction of new web features and technologies, making it an interesting read. Specifically, the paper covers modern web technologies such as Service Workers, Edge Workers, Wasm, and advanced client-side storage mechanisms, among others, and their impact on the attack surface. This breadth is valuable in understanding the emerging security risks.

* The criticism against the OWASP XSS taxonomy seems to hold merit, as the traditional taxonomy cannot accurately capture all the potential attack scenarios involving new web technologies. On the other hand, however, the new taxonomy proposed in this paper does not seem to constitute an optimal approach as it is very detailed and broad. With the inclusion of execution environments, contexts, and storage mechanisms, as well as the notion of a continuum for expressing persistence, it risks becoming overly broad as new technologies emerge. Furthermore, while the paper provides a level of precedence and persistence for existing execution environments, contexts, and storage mechanisms in Figure 3, these are not universally agreed upon. This broadness, although it provides flexibility, might confuse developers and other relevant parties, ultimately limiting the utility of the taxonomy.

* Additionally, the new taxonomy can be seen as an incremental improvement over the OWASP taxonomy rather than a fundamentally new approach. The focus on "infection" and "affection" feels like a rephrasing of existing ideas with added flexibility, which could also lead to broadness and loose interpretation.

* It seems that the paper does not make significant contributions beyond proposing this new taxonomy, which is its main contribution. The rest of the paper presents various attack vectors for the different new web technologies and storage mechanisms, but this part reads more like a systemization of knowledge. Even the attack based on the Service-Worker-Allowed header presented in Section 6 has been known for some time. I suggest that the paper’s contributions be clearly outlined in the introduction section.

* The paper could be improved significantly if the potential attack vectors discussed had been implemented and assessed experimentally, demonstrating their effectiveness. Another suggestion would be an exploration of the prevalence of such vectors on the Web or identifying already disclosed vulnerabilities (CVEs) and classifying them according to the proposed taxonomy, which could significantly improve the paper.

* Figure 5 is very important, and should be moved in the main body of the paper.

**Questions:**

Please see comment 2

**Reviewer Confidence:**

3: The reviewer is confident but not certain that the evaluation is correct

**Scope:**

4: The work is relevant to the Web and to the track, and is of broad interest to the community

---

### Official Review · Reviewer_GLPB · 2024-12-03

**Novelty:** 5
**Technical Quality:** 4

**Review:**

The paper titled "MatriXSSed: A New Taxonomy for XSS in the Modern Web" presents an ambitious attempt to redefine how cross-site scripting (XSS) is conceptualized in the context of modern web technologies. The authors argue that the traditional taxonomy of XSS (persistent, non-persistent, server-side, client-side) is inadequate for describing the complexities introduced by contemporary web features such as service workers, web assembly, and various client-side storage mechanisms. They propose a more nuanced framework that emphasizes code "infection" and "affection" stages, considering persistence as a continuum rather than a binary attribute.

While the paper introduces interesting ideas and showcases critical insights, several areas require improvement.

Pros:
Novel Contribution: The paper presents a bold rethinking of the XSS taxonomy, which is necessary given the technological advancements in web applications. By moving beyond the traditional client-server dichotomy, the authors address gaps in the current understanding of XSS, particularly when involving service workers, web assembly, and new storage APIs.
Clarity in Structure: The paper is well-organized, with each section logically building upon the last. The theoretical discussion of XSS is followed by a detailed proposal for a new framework, and the paper concludes with practical demonstrations of vulnerabilities that exploit new web features.
Relevance: The topic is highly relevant to the WebConf community, given the increasing complexity of web security. Cross-site scripting remains one of the most pervasive attack vectors, and addressing the limitations of the current taxonomy is a significant step forward for web security research.

Cons:
Lack of Concrete Validation: Although the paper presents theoretical arguments and demonstrates examples of modern XSS vectors, it lacks a rigorous experimental evaluation. A more thorough quantitative analysis comparing the proposed taxonomy's effectiveness to the traditional one would strengthen the paper's claims.
Limited Discussion on Defenses: The section on defenses against XSS does not offer substantial new insights beyond what is already known. The authors acknowledge the limitations of existing mitigation techniques, but they do not provide a clear path forward. A deeper exploration of how their taxonomy can inspire new defense mechanisms would make the contribution more impactful.
Terminology Overload: The introduction of new terminology such as "infection," "affection," and persistence as a "continuum" is intriguing, but these concepts are not sufficiently grounded in practical examples. The paper would benefit from more real-world scenarios to illustrate these ideas and make them more accessible to practitioners.
Comparative Analysis Missing: The paper mentions prior work in passing but does not provide a thorough comparative analysis with existing taxonomies of XSS. A more explicit discussion comparing the benefits and limitations of the new framework versus the traditional matrix would enhance the paper's clarity and relevance.

**Questions:**

Questions for the Authors:

Could you clarify how the proposed taxonomy applies to legacy web technologies or systems? Is the new framework exclusively relevant to modern web applications?
Have you conducted any empirical evaluations to validate the effectiveness of the proposed taxonomy? If so, could you share the results, or if not, how do you plan to assess its practical impact?
The "continuum of persistence" is an interesting concept, but how would you propose integrating this into existing automated tools for XSS detection or mitigation?
Could you provide more examples of how the proposed taxonomy improves the identification or mitigation of XSS compared to the traditional client/server-side matrix?

**Reviewer Confidence:**

3: The reviewer is confident but not certain that the evaluation is correct

**Scope:**

4: The work is relevant to the Web and to the track, and is of broad interest to the community